



# Simple algorithms to compute meridional overturning and barotropic streamfunction on unstructured meshes

Dmitry Sidorenko[1], Sergey Danilov[1,2,3], Nikolay Koldunov[1,4], Patrick Scholz[1], and Qiang Wang[1]

[1]Alfred Wegener Institute for Polar and Marine Research, Bremerhaven, Germany
[2]Jacobs University, Bremen, Germany
[3]A. M. Obukhov Institute of Atmospheric Physics RAS, Moscow, Russia
[4]MARUM – Center for Marine Environmental Sciences, Bremen, Germany

**Correspondence:** D. Sidorenko (dmitry.sidorenko@awi.de)

**Abstract.** Computation of barotropic and meridional overturning streamfunctions for models formulated on unstructured meshes is commonly preceded by interpolation to a regular mesh. This operation destroys the original conservation which can be then artificially imposed to make the computation possible. An elementary method is proposed that avoids interpolation and preserves conservation in a strict model sense. The method is described as applied to the discretization of the Finite volumE

Sea ice – Ocean Model (FESOM2) on triangular meshes. It however is generalizable to collocated vertex based discretization on triangular meshes and to both triangular and hexagonal C-grid discretizations.

## 1   Introduction

Over recent years a considerable progress has been achieved in the development of global ocean circulation models working on horizontally unstructured meshes such as FESOM1.4, Wang et al. (2014), MPAS-o Ringler et al. (2013), FESOM2, Danilov

et al. (2017) and ICON-o Korn (2017). By refining in dedicated areas of the world ocean these models may resolve dynamics that would otherwise require nesting or using higher resolution globally. Since these models still use vertically aligned meshes the overhead of horizontally unstructured mesh is minimized because the horizontal neighborhood information is valid for the entire vertical column, and becomes negligible as the number of vertical levels is increased. These models show a very good parallel scalability and reach throughput (in simulated years per day) comparable to that of structured-mesh models (Koldunov

et al. (2019)). However, the unstructured character of meshes makes many traditional diagnostics, such as barotropic and meridional overturning streamfunctions, difficult. Any interpolation on a regular mesh violates the sense in which continuity is satisfied in a model and introduces errors which, while often acceptable for computing local fluxes and transports, are very annoying in computations of global or basin streamfunctions where large positive and negative contributions are combined together. Furthermore, in the case of streamfunctions one is most frequently interested in variability, which might be easily

masked or biased by the inconsistencies introduced by the analysis procedure. In the early version of FESOM, based on continuous finite elements, the situation was exacerbated by continuity being formulated in a weighted sense, without explicitly computed fluxes (Sidorenko et al. (2009)).





All new large-scale ocean models are based on the finite-volume method and as such have a clear definition of fluxes at boundaries of the control cells of their meshes. However, these fluxes are defined on irregularly located faces, so instead of using them in their original sense one is tempted to rely on interpolation to a regular mesh. Our practice shows that incurring inconsistencies can be large, and this road should not be followed if global or basin-scale quantities are computed. It turns out that there are efficient and easy to implement procedures that are based on exact fluxes and balances and might be used for analyses. These procedures do not rely on interpolation, but use binning, which is sufficient in most cases except for very coarse meshes. The intention of this note is to describe some of them. In doing so we will use the arrangement of variables of FESOM2, however the adjustments needed for other models with different discretizations are relatively straightforward and will be briefly mentioned. We suspect that similar procedures are already used by other groups (in particular, for the analysis on cubed-sphere meshes of the Massachusetts Institute of Technology general circulation model see eg. Adcroft et al. (2004)), but we feel that they need to be documented for unstructured meshes, facilitating the use of unstructured-mesh models by a broader community.

We will discuss computations of meridional overturning streamfunction in height and density coordinates as well as computations of barotropic streamfunction.

## 2 Geometry of discretization

FESOM2 uses a cell-vertex discretization, placing horizontal velocities on centroids of triangles and scalar quantities at vertices if viewed from the surface, as shown schematically in Fig. 1. These quantities are stored at midlevels. Vertical velocities are located at vertices and full levels. We use index $v$ to enumerate vertices, $c$ (cells) to enumerate triangles and $k$ to enumerate vertical levels or midlevels (centers of layers). The velocity control volumes are mesh triangles, and scalars are associated with median-dual control volumes formed in the horizontal plane by connecting midpoints of edges with cell centroids. On uniform equilateral meshes they coincide with hexagons of dual mesh, but they generally differ. For the reasons discussed in Danilov et al. (2017) the bottom topography of FESOM is given on cells, implying that velocity control volumes are triangular prisms in 3D. However, a part of scalar control volume can be cut by bottom topography at depths, and its footprint will differ fro that at the surface. As a consequence, there is a one-dimensional array $A_c$ of triangle areas, and a two-dimensional array $A_{kv}$ of the areas of scalar control volumes. The transport through the top face of scalar prism with indices $(k, v)$ is $w_{kv}A_{kv}$, with $w_{kv}$ the respective vertical (or cross-level in the case of moving level surfaces) velocity. Each triangle is characterized by the list of its vertices $v(c)$ which is $(v_1, v_2, v_3)$ for $c = c_1$ in Fig. 1.

The elementary structure used in computations of horizontal fluxes between two scalar control volumes is given by mesh edge (labelled with index $e$). An edge is characterized by its two vertices $(v_1, v_2)$ symbolically written as $v(e)$, and two cells it belongs to, $(c_1, c_2)$ symbolically written as $c(e)$. For boundary edges $c_2$ is absent, and $c_1$ is the left cell to the edge direction, which is from the edge first vertex to the second one. There are two vectors drawn from edge midpoint to centroids of edge cells, $(\mathbf{d}_{ec_1}, \mathbf{d}_{ec_2})$. Their components are expressed in local Cartesian coordinates related to respective cells. The transport

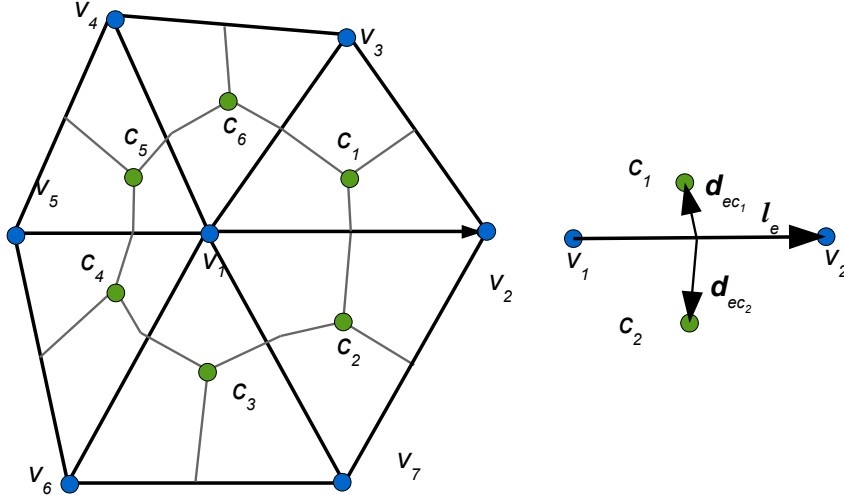

**Figure 1.** Horizontal schematic of median-dual control volumes (left) and the edge-based structure (right). In FESOM2, scalar quantities and vertical velocity are at vertices (blue circles), while the horizontal velocities are at triangle centroids (green circles). The median-dual control volume around vertex $v_1$ is bounded by segments (gray lines) connecting the centers of neighbor triangles with midpoints of edges. Edge $e$ (right panel) is characterized by its vertices $v(e) = (v_1, v_2)$ and cells $c(e) = (c_1, c_2)$ with $c_1$ on the left. The edge vector $\mathbf{l}_e$ connects vertex $v_1$ to vertex $v_2$. The edge cross-vectors $\mathbf{d}_{ec_1}$ and $\mathbf{d}_{ec_2}$ connect the edge midpoint to the respective cell centers.

through the faces of scalar control volume in layer $k$ in the direction of edge is

$$F_e = [-(\mathbf{e}_z \times \mathbf{d}_{ec_1}) \cdot \mathbf{u}_{kc_1} h_{kc_1} + (\mathbf{e}_z \times \mathbf{d}_{ec_2}) \cdot \mathbf{u}_{kc_2} h_{kc_2}] T_e,$$

where $\mathbf{e}_z$ is a unit vertical vector, $h_{kc_1}$ and $h_{kc_2}$ are the layer thicknesses at respective velocity points and $T_e$ is the tracer estimate at edge midpoint. $T_e = 1$ for volume transport. In MPAS-o or ICON-o codes, which are based on hexagonal and triangular C-grid discretizations, normal velocities are located at edges and computations of transports are simpler. The arrangement of hexagonal C-grid is easily obtained from the case considered here if edges of dual triangular mesh are considered (with the difference that centroids are replaced by circumcenters and line connecting $c_1$ with $c_2$ are perpendicular to edge $e$. Importantly, edge-related transports are the same as in model, however a care should be taken that $T_e$ is computed in the same way as in the model if property fluxes are analysed.

## 3 Meridional overturning

There are two convenient ways of computing meridional overturning in geopotential coordinates on unstructured meshes. The first one involves vertical velocities. It is more straightforward and, as we guess, generally known. The second one is based





on horizontal velocities. It is slightly more complicated, but allows generalization to isopycnal coordinates. The meridional overturning streamfunction $\Psi(z,\theta)$ is defined as

$$\Psi(z,\theta) = \int\limits_{\theta_s}^{\theta} R_E d\theta \int\limits_{x_w}^{x_e} w dx,$$

or

$$\Psi(z,\theta) = \int\limits_{-H}^{z} dz \int\limits_{x_w}^{x_e} v dx.$$

In this definitions $v$ is the meridional velocity component, $H$ the ocean bottom depth, $x_w$ and $x_e$ the western and eastern boundaries in zonal direction, $\theta$ and $\theta_s$ are the latitude and the southern latitude, and $R_E$ Earth's radius. These definitions are equivalent because full velocity vector is divergence-free.

### 3.1 Method A

In FESOM2, the vertical velocity is conservatively remapped from vertices to cells using

$$w_{kc} = \sum_{v \in v(c)} w_{kv}/3, \quad k \neq N_c, \quad w_{N_c c} = 0,$$

where $v(c)$ is the list of vertices of triangle $c$ and $N_c$ is the number of the bottom level on triangle $c$. Indeed, it is easy to prove that $\sum_v A_{kv} w_{kv} = \sum_c A_{kc} w_{kc}$ for FESOM2 discretization, so that the vertical (across level surface) transport is preserved. Using triangles is more convenient in FESOM2 because bottom depth is constant on triangles. This remapping is not required in ICON-o and MPAS-o where the bottom depth is specified at scalar locations.

We introduce a set of latitude bins $(\theta_i, \theta_{i+1})$, $\theta_i = \theta_0 + i\Delta\theta$, $i = 0, \ldots, N_\theta$ covering the ocean domain. The procedure of computations is straightforward and is illustrated schematically in Fig. 2.

- For each bin $i$ find the list of triangles $c(i)$ with centroids in these bins. They will be partly masked by bottom topography in deep layers, and we will formally write this list as $c(ki)$, adding a layer index $k$. Subsequent computations are over triangles and levels, so that only $c(ki)$ is needed.

- Compute $\Delta\Psi_{ki}$ as

$$\Delta\Psi_{ki} = \sum_{c \in c_{ki}} w_{kc} A_c,$$

where $c_{ki}$ is the list of triangles the centers of which are in bin $i$ at level $k$.

- Compute the meridional overturning streamfunction

$$\psi_{ki} = \sum_{j=1}^{i} \Delta\Psi_{kj}.$$





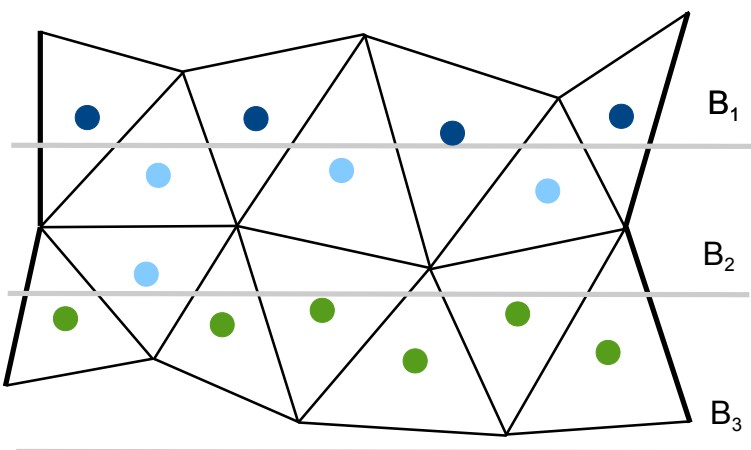

**Figure 2.** Schematics of binning. Circles correspond to triangle centroids. Bins (here $B_1$, $B_2$ and $B_3$) are given by selected latitude lines. A triangle is in a bin if its centroid is in this bin. Triangles with centroids in dark blue, light blue and green fit in bins $B_1$, $B_2$ and $B_3$ respectively.

The procedure as written is strictly applicable in the case when level surfaces are fixed except for the surface. For $z^*$ vertical coordinates or for other options where where level surfaces are changing only slightly around their mean positions it can still be used in most cases. It can be readily augmented with vertical remap to fixed levels by considering that the difference in transports $(w_{kc} - w_{(k+1)c})A_c$ is linearly distributed within the layer in case when layers do not disappear, and level surfaces
5   do not outcrop and stay at fixed depths where they cross topography. The method B should be used in more general case.

Generally $\Delta\theta$ should be taken about or larger than the typical size of triangles. The triangles that are counted as belonging to a bin are not necessarily confined to this bin, and the total area occupied by them differs from the bin area. However, there generally are sufficiently many triangles in each bin, and one gets a smooth $\Psi_{kj}$ despite these effects. The procedure can be improved by conservative remapping into bins, which might be needed on coarse meshes. One may always check the
10  bin attribution effect by repeating computations with smaller $\Delta\theta$. We also note that for instantaneous vertical velocities the procedure may result in $\Psi$ different from zero at the surface. It will become zero only upon sufficient averaging which removes transient behavior of the surface.

The computations presented here can be generalized to some other sets of binning. Any sufficiently smooth scalar quantity defined at vertices or triangles can be used to introduce a set of bins. For example, being limited to the NA subpolar gyre, one
15  may ask where the AMOC is forming using bins in mean sea surface height or barotropic streamfunction (see, e.g., Katsman et al. (2018)).

In the following we present an example showing differences between computations using different bins in $\Delta\theta$. For this, FESOM was configured on a mesh with resolution varying from nominal one degree in the interior of the ocean to ~1/3 degree

in the equatorial belt and ~24 km north of 50°N. We run the model for one year starting from climatology and compute the MOC from the annually averaged velocity. Because of starting the model at rest and short period of averaging we expect $\partial\eta/\partial t \neq 0$, where $\eta$ is the sea surface height. This, however, shall not affect the presented results. Fig. 3 depicts the simulated global MOC which is expressed by the basinwide mid-depth cell of ~20Sv at 40°N and the bottom cell, induced by the circulation of the

5 Antarctic Bottom water with a maximum of 10Sv. Bins with $\Delta\theta = 0.125\,°$, which are finer than the nominal resolution, have been used for computing the streamfunction. Differences between computations using different bins in $\Delta\theta$ are shown in Fig. 4. Using the the coarsest bin size of $4\,°$ the difference in MOC reaches locally above 5 Sv. As one would expect, decreasing the size of bins leads to convergence towards the solution obtained with the finest bin size of $\Delta\theta = 0.125\,°$. We see that using bins of $\Delta\theta = 0.25\,°$ is already sufficient in this case because the mesh contains only few triangles that are smaller than the bin size.

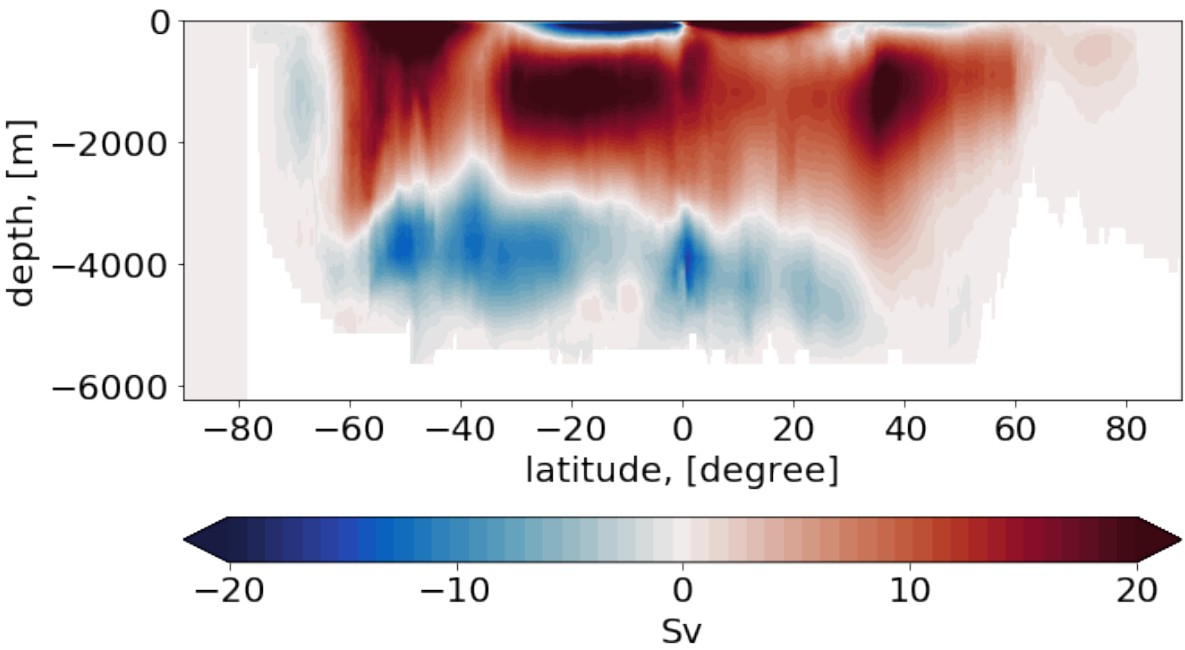

**Figure 3.** Global Meridional Overturning Circulation (MOC) streamfunctions including the eddy induced transports. The streamfunctions depicts a canonical pattern as known from the literature with a maximum of 20Sv at 45°N.

10 ## 3.2 Method B

Here the horizontal velocities are used. We select a set of latitudes $\theta_i$. The steps of the procedure are as follows.

- For each $i$ draw a line $\theta = \theta_i$ and find a set of edges crossed by this line, as shown schematically in Fig. 6. For this, cycle through all edges, picking up those that satisfy the condition $(\theta_{v_1} - \theta_i)(\theta_{v_2} - \theta_i) < 0$, with $\theta_{v_1}$ and $\theta_{v_2}$ the latitude of edge vertices. To avoid situations when the line passes exactly through the mesh vertex, a random noise of small





amplitude is added to the original $\theta_i$ before edge $e$ with vertices $(v_1, v_2)$ is tested. Schematic in Fig. 6 shows that the actual line through which transport is computed is a broken line composed of vectors $(\mathbf{d}_{ec_1}, \mathbf{d}_{ec_2})$ related to the crossed edges. For a triangular C grid discretization one will deal with transports directly through the edges. On hexagonal C grids the procedure needs to deal with edges of dual triangular mesh. We denote the list of edges intersected by the line

$\theta = \theta_i$ as $e(i)$.

– The flux associated to the edge is given by the expression for $F_e$ above. The question now is the orientation of edges. This question is trivially solved for each $e$ by taking $F_e$ if $\theta_{v_1} - \theta_i > 0$ and $-F_e$ otherwise. It corresponds to keeping the normals to segments oriented so that transports are from the "northern" side of the broken curve. On triangular C grid the edge normal vectors used to introduce edge velocities can be selected as turned $90°$ in positive direction from the

edge direction. This will allow to solve the orientation problem in the same way.

– Since each of segments $(\mathbf{d}_{ec_1}, \mathbf{d}_{ec_2})$ belongs to a particular cell, vertical integration is trivial for fixed level surfaces. If level surfaces are moving, the fluxes (transports) through the faces associated with segments are conservatively interpolated to the desired system of levels assuming linear distribution within model layers. In particular, the new system of levels can be specified in terms of potential density, with the result being the streamfunction in density coordinates. For

each level the contributions from edges $e \in e(i)$ are summed to get streamfunction at this level and the latitude $\theta_i$.

Note that the list of intersected edges may be ordered arbitrarily, the computation relies on the orientation of edges with respect to lines $\theta = \theta_i$. This is the reason why the search for intersected edges remains relatively fast even on very large meshes. Furthermore, it needs to be done only once for a particular mesh. Similarly to Method A, computations can be generalized to any set of lines, in particular to isolines of mean sea surface height or barotropic streamfunction. In both Methods we introduce

masks if computations need to be confined to a particular basin.

Using this method we computed the streamfunction using the discrete spacing of $\Delta\theta = 0.125°$. The difference to the streamfunction computed by method A is illustrated in Fig. 7. The discrepancy between both methods is caused by the difference of attribution of ocean volume to $\theta_i$. This, as shown in Fig. 7, can lead to a differences exceeding locally 1 Sv. These differences are not the errors, but uncertainty in the interpretation (see further).

As has been mentioned above the advantage of method B is the possibility of computing the MOC for a new system of vertical levels. Figure 5 depicts the MOC computed using $\sigma_2$ (density referenced to 2000m) coordinate in vertical. For computing the streamfunction in density coordinate we used 1000 equally spaced $\sigma_2$ levels varying from 1027.5 to 1037.5 $kg/m^3$. The resulting MOC resembles that of generally known pattern from literature with less expressed Deacon cell as if $z$ coordinate is used. The result is sensitive to the selection of density bins, as illustrated in the bottom panel of Fig. 5 where the difference

is presented with computations relying on the density levels of Megann (2018). He used 72 unequally spaced density classes spanning the range $30.0 < \sigma_2 < 37.2 \ kg/m^3$ and using the logarithmic scale for densities higher than $\sigma_2 > 35.0 \ kg/m^3$ to better represent the deep and bottom waters. Thus, due to the different sampling the difference in the equatorial overturning of the surface waters reaches ~3Sv for $30 < \sigma_2 < 35.0 \ kg/m^3$ and is even larger for the circulation cell associated with the Antarctic Bottom Water. We conclude that different or not detailed enough selection of density levels may result in the small-





scale recirculations in diagnostic of the MOC. However, this difference is not an error but attribution uncertainty created by arbitrariness in the selection of density levels.

## 4 Barotropic streamfunction

As follows from the equation for elevation, time mean vertically integrated horizontal velocity $\mathbf{U}$ is divergence free, $\nabla \int_{-H}^{\eta} \mathbf{u} dz = \nabla \mathbf{U} = 0$, i.e. it can be written in terms of the barotropic streamfunction $\overline{\Psi}$ as

$$\mathbf{U} = -\nabla \times (\overline{\Psi} \mathbf{e}_z).$$

This streamfunction gives vertically integrated transport between two points at the surface.

### 4.1 Computations through binning

The barotropic streamfunction is more difficult to compute because binning has to be done in two directions. We introduce first a set of lines $\phi = \phi_j$, where $\phi$ is the longitude, and $\phi_j$ is the set of equally spaced longitude values over the basin of interest. As a first step the set of broken lines associated to each straight line $\phi = \phi_j$ is found. As the next step vertically integrated transports associated with the segments of broken line are computed. The final step is further binning of edges and associated transports into equally spaced latitude intervals $(\theta_i, \theta_{i+1})$. Transport (and hence streamfunction) at each bin can be then computed by summing contributions going from the southern boundary where $\overline{\Psi}$ is set to zero.

This procedure can potentially be more noisy than computations of MOC, and may benefit from a conservative remap of the contributions from the segments in the second binning step (the number of segments in final bins is not necessarily large, in contrast to computations of meridional overturning).

According to the above procedure we computed the barotropic streamfunction using $\Delta\theta, \Delta\phi = 0.25°$. Considering, that the procedure requires two-fold loop for $(\Delta\theta_i, \Delta\phi_j)$ in case of large meshes and small bins it can become computationally heavy. The result is illustrated in Fig. 8 (upper panel) and depicts reasonable structure of the main gyres with transports of 160 Sv and 70 Sv across Antarctic Circumpolar Current (ACC) and Gulf Stream, respectively.

In Fig. 8, middle and bottom panels show the differences between the streamfunctions if bins of $2°$ and $1°$, respectively, are used. As expected, the largest differences occur along the main gradients and reach of above 5 Sv along the ACC front. As in case with the MOC we note that these differences are not the errors, but uncertainty created by arbitrariness in the selection of bin size.

### 4.2 Computations through velocity curl

FESOM2 as its predecessor use implicit time stepping for the internal mode. The already available solver and routines need to be only slightly adjusted to compute the barotropic streamfunction $\overline{\Psi}$ in the case when no-slip boundary conditions are applied. Taking curl of the equation defining $\overline{\Psi}$ one gets

$$\Delta\overline{\Psi} = \zeta, \quad \zeta = \mathbf{e}_z \cdot \nabla \times \mathbf{U}.$$





In FESOM the discrete $\zeta$ is located at scalar points (at vertices), so modifications of the sea surface height solver to solve the above equations are indeed elementary. The difficulty in formal application of this approach is that the equation above needs to be solved in a multiply connected domain with the Dirichlet boundary conditions provided on the periphery of each island and continent. Although these conditions can be formally provided by drawing lines connecting the islands and computing

transports through the associated broken lines, this is tedious enough, especially when mesh resolution is high (and there are many islands). In the case of no-slip boundary conditions circulations along each island are identically zeros, and the equation above can be formally solved with the Dirichlet boundary condition on the southern boundary and the von Neumann boundary condition $\partial \overline{\Psi}/\partial \mathbf{n} = 0$ ($\mathbf{n}$ is the normal to the boundary). Although this condition does not ensure that $\overline{\Psi} = \mathrm{const}$ over the periphery of any island, our experience with FESOM1.4 is that it works fine enough for practical purposes.

If we integrate the equation above over a scalar control volume (in FESOM2 scalar points are natural locations for relative vorticity $\zeta$ and streamfunction), we get

$$\sum_{e=e(v)} [-(\mathbf{e}_z \times \mathbf{d}_{ec_1}) \cdot \nabla \overline{\Psi}_{kc_1} + (\mathbf{e}_z \times \mathbf{d}_{ec_2}) \cdot \nabla \overline{\Psi}_{kc_2}] = \sum_{e=e(v)} [\mathbf{d}_{ec_1} \cdot \mathbf{U}_{c1} - \mathbf{d}_{ec_2} \cdot \mathbf{U}_{c2}].$$

The contributions from edges on boundaries here are one-sided, including only segments that are wet (the first in the list in the case of FESOM). This automatically takes into account that there are no contributions from the boundary, as is the case for the no-slip boundary conditions. The operator on the left hand side in the case of FESOM is, up to the absence of depth weighting, the same as the part of operator used to compute the elevation, so the implementations is straightforward in the code (less so for post-processing). A clear drawback of this procedure is that it is not applicable for partial slip boundary conditions (it can be

generalized, but will become too complicated). Since the methods based on bins was found to perform reliably, the curl-based method presents largely a historical interest.

## 5   Technical realization

The FESOM 2.0 source code is available at https://github.com/FESOM/fesom2 . It is written in Fortran 90 with some C/C++ code for providing bindings to some of the third party libraries. The code employs the distributed memory parallelization

based on MPI to run on HPC systems. The presented diagnostics have been computed using python routines that are part of the FESOM 2.0 code distribution. For computing the MOC in $z$ coordinate python routines require velocities to be stored as $(u, v)_{k,c}$ where $k$ and $c$ denote the layer and element indices. This is the default output provided by FESOM. For computing the barotropic streamfunction and MOC in density space the index $k$ refers to a density bin and $(u, v)_{k,c}$ denotes the transport through this bin below the element $c$. Transports within the density classes are instantaneously computed by FESOM and

stored with the desired frequency if option $ldiag\_dMOC$ is activated. For the sake of better subsampling, the number of density classes for computing transports shall be sufficiently large. This, however, can make the remapping of transports onto density bins computationally heavy. For this reason $ldiag\_dMOC$ is switched off per default.





## 6 Discussion

The general idea of simple procedures described above is the use of transports as they are defined in an unstructured-mesh model, avoiding interpolation from an unstructured to a structured mesh. The diagnosed quantities such as meridional and barotropic streamfunctions rely on the continuity equation, which is satisfied by the model only in a certain discrete sense.

Interpolation destroys this sense, requiring corrections and introducing interpretation errors related to these corrections. In practice the interpretation errors are significant, being on the level of Sverdrups for the meridional overturning as illustrated in Sidorenko et al. (2009), hampering discussions of MOC variability.

The algorithms above rely only on transports as they defined in models, and use conservative interpolation only in the vertical direction if required by a specified system of levels.

We emphasize that algorithms described above still contain interpretation uncertainty, for in each case there is some sensitivity to how bins or vertical levels are selected. In Method B the straight line $\theta = \theta_i$ can be considered as centered in the respective bin, however the broken line drawn around the straight line is not necessarily centered within a bin. Drawing other possible broken lines in the bin is generally possible and can be proposed to estimate this uncertainty. However, we would argue that such uncertainty is intrinsic in the notions we are willing to diagnose: they must rely on transport strictly consistent

with model discretization to avoid errors, and such transports are defined at irregular locations that generally do not lie on lines of latitude or longitude. A set of bins proposes some interpretation of integrated transports that is free of horizontal interpolation. Any attempt to interpolate may create new uncertainties instead of making the analysis more accurate. These 'attribution' uncertainties have to be kept in mind especially in situations where small variability of MOC is the subject of analysis. Our experience thus far with the methods described above is that the computed patterns of MOC and barotropic streamfunction are

sufficiently smooth.

## 7 Conclusions

We describe a set of simple procedures intended to diagnose the meridional overturning and barotropic streamfunctions intended for unstructured meshes and requiring no interpolation of model output to regular meshes. We give application examples and discuss uncertainties involved. The procedures are described for FESOM2, but their adaptation for other discretizations

(MPAS or ICON) is straightforward. Our experience with using them indicates that they create much less difficulties with interpretation of model results than all our previous approaches based on interpolation.

*Code availability.* The code of the FESOM 2.0 model which was used to conduct the simulations for this paper is available at Zenodo (Sidorenko et al., 2020). The latest version of FESOM2 code can be downloaded from the public GitHub repository at https://github.com/FESOM/fesom2 under the GNU General Public License (GPLv2.0).



*Data availability.* Dataset related to this article can be found at Zenodo (Sidorenko et al., 2019)

*Author contributions.* DS and SD proposed the methods. DS and NK worked on the implementation, and NK, PS and QW contributed to the analyses. DS and SD wrote the the initial manuscript and all authors contributed to its final version.

*Competing interests.* The authors declare that they have no conflict of interest.

5  *Acknowledgements.* This paper is a contribution to the projects S1 (Diagnosis and Metrics in Climate Models) and S2 (Improved parameterisations and numerics in climate models) of the Collaborative Research Centre TRR 181 "Energy Transfer in Atmosphere and Ocean" funded by the Deutsche Forschungsgemeinschaft (DFG, German Research Foundation) - Projektnummer 274762653. D. Sidorenko and Q. Wang are funded by the Helmholtz Climate Initiative REKLIM (Regional Climate Change). The runs were performed at the AWI Computing Centre.





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



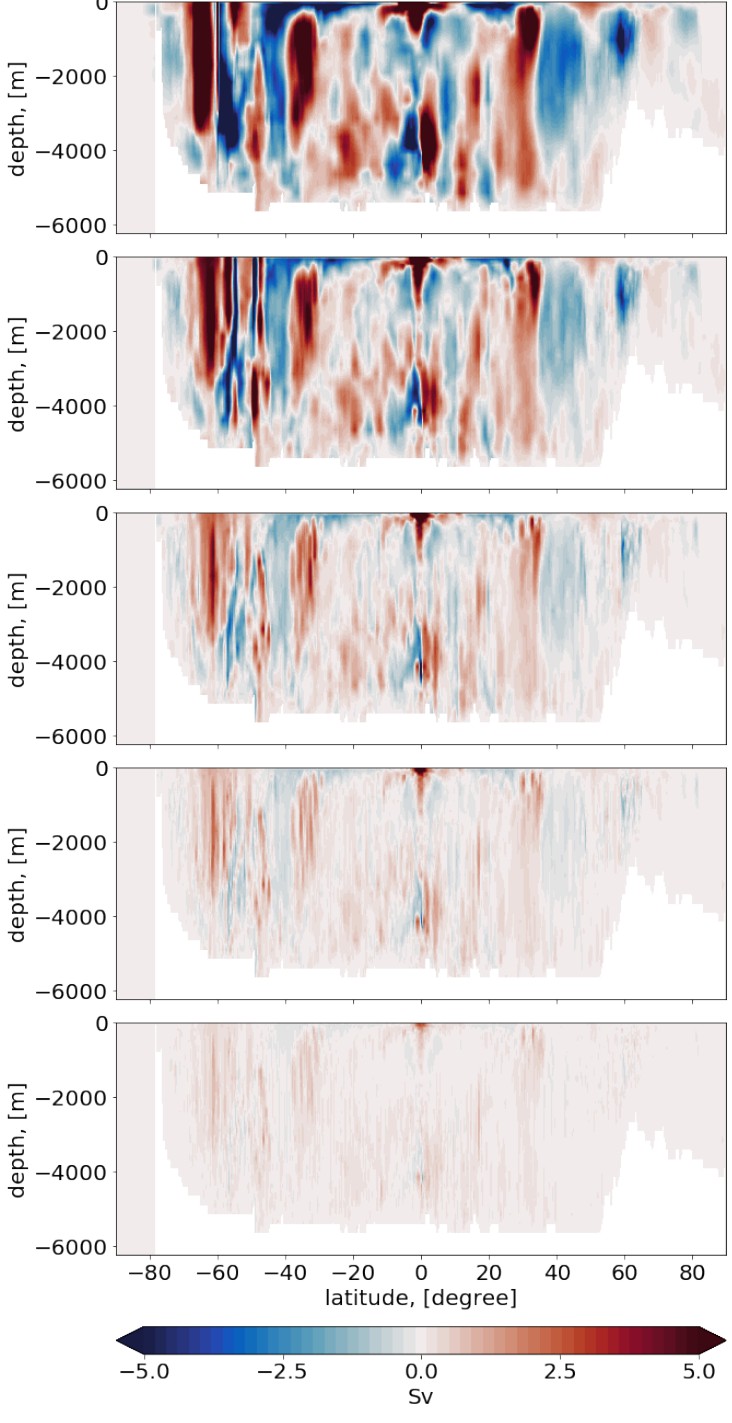

**Figure 4.** Differences in MOC computed with bins $\Delta\theta = 4°$, $2°$, $1°$, $0.5°$, $0.25°$ (from top to the bottom) relative the MOC computed with $\Delta\theta = 0.125°$. Evidently there is a convergence with decreasing bin size.



**Figure 5.** upper panel shows the MOC computed using 1000 equally spaced density levels varying from 1027.5 to 1037.5 $kg/m^3$. Lower panel shows the difference in MOC if 72 unequally spaced vertical levels after Megann et al. 2010 are used.



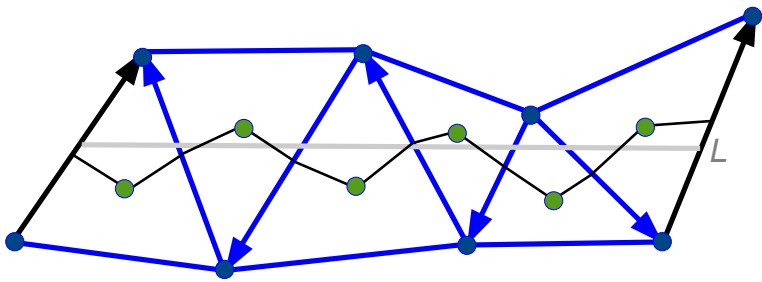

**Figure 6.** Schematics of edge search method. The gray line $L$ intersects edges depicted with arrows that show their orientation. The set of segments drawn to centroids from the centers of intersected edges forms a broken line connecting land at left to land at right where exact expressions for fluxes are available in FESOM2. The broken line formed by the intersected edges will be taken on triangular C grids, and on hexagonal C grids it will be composed of edges of primary hexagonal mesh. The set of intersected edges may stay disordered, only edge orientation with respect to the line $L$ should be known. The latter is positive if the latitude of the first edge point is larger than that of $L$ and negative otherwise. The transport through $L$ is the transport through the associated broken line.

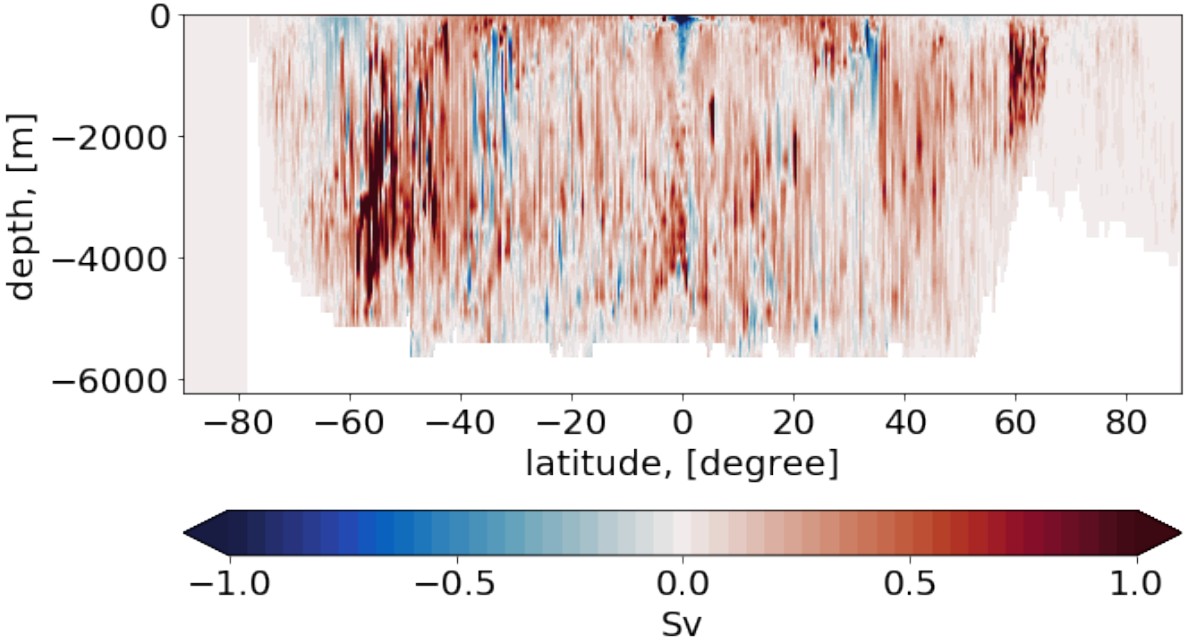

**Figure 7.** Differences between computations of MOC using meridional or vertical velocities. The discrepancy between techniques may result in differences of more than 1 Sv.

**Figure 8.** Uppper panel: the barotropic streamfunction computed using $\Delta\theta, \Delta\phi = 0.25°$. Middle and bottom panels show the differences in cases $\Delta\theta = 1°$ and $2°$, respectively.