# Peer review of "Simple algorithms to compute meridional overturning and barotropic streamfunction on unstructured meshes"

_Geoscientific Model Development, 2019_

## Short Comment (SC1) · 11 May 2020

Please see attached pdf for a better formatted version of these comments.

This paper provides a clear explanation on how to compute the meridional overturning and barotropic streamfunction on unstructured meshes. It is well-written and clearly communicates the motivation, the mesh structure, and mathematics of the computation, and the sensitivity of the parameters. I have not seen these details discussed in other publications, so it will be a useful reference to ocean modelers reproducing this calculation.

[Figure]

In MPAS-Ocean, we compute the meridional overturning circulation (MOC) streamfunction by integrating the vertical velocity in latitude bins, level-wise, as described in this paper as Method A. I adapted this method from our previous model, POP, which is a structured quadrilateral mesh but usually ran on a tri-pole grid, so the generality of latitude binning was required for that model as well. We have the ability to run the MOC calculation as an in-situ calculation during the simulation: https://github.com/MPAS-Dev/MPAS-Model/blob/master/src/core_ocean/analysis_members/mpas_ocn_moc_streamfunction.F#L518 or as a post-processing calculation from the time-averaged velocity fields: https://github.com/MPAS-Dev/MPAS-Analysis/blob/develop/mpas_analysis/ocean/streamfunction_moc.py#L436 We describe the philosophy of the in-situ computation here: Woodring, J., Petersen, M., Schmeisser, A., Patchett, J., Ahrens, J., Hagen, H., In Situ Eddy Analysis in a High-Resolution Ocean Climate Model, IEEE Transactions on Visualization and Computer Graphics, vol.22, no.1, pp.857-866, Jan. 31 2016 http://ieeexplore.ieee.org/stamp/stamp.jsp?arnumber=7192723 If you feel it is relevant, you are free to add any of that background to your paper.

We have found that the python post-processing method is fine at lower resolutions, but at high resolutions ($\sim$ 1 million horizontal cells) it is extremely slow or fails, and we rely on our in-situ calculation. It would be good to hear in section 5 about any performance limitations in your calculation. On page 9 line 20, are these python routines just the plotting part, or do they reproduce the functionality of the Fortran and c code? If the later, did you run into performance bottlenecks, and how did you run python at high resolution – some combination of xarray and dask?

Your equations are missing numbers.

I find the notation of the streamfunction definition confusing (page 4, top). I prefer to see the d\theta and dz at the very end, to indicate what is included in the integral. The first one, in particular, could be confused with the multiplication of two integrations,

rather than one case of a double integral. Another way to clarify is to write w and v as: w(x,\theta',z) and v(x,\theta,z') and integrate in \theta' on the first equation and z' on the second, to explicitly state that the primes are dummy integration variables, not to be confused with the upper bound and independent variable on the left side.

For the new reader, it would be useful to review the definition of a streamfunction at the beginning of section 3. That is, if we have zonally-integrated velocities W and V, then for an incompressible fluid we can define a streamfunction \Psi as d\Psi/d\theta = W (1) d\Psi/dz = -V (2) and refer to some fluid dynamics textbook – my favorite is Kundu, https://www.amazon.com/Fluid-Mechanics-Pijush-K-Kundu/dp/0123821002 (I forget which line is negative, I always have to work through it. I assume one of your streamfunctions has a negative sign absorbed in a coordinate or variable definition, but that would be good to note as well).

From my equations above, it is clear that one may compute \Psi by either integrating (1) in \theta, or integrating (2) in z. This is identical analytically, but leads to different implementations and numerical results, as you describe.

Please state the boundary conditions for the two equations, top of page 4. For Method A and a global MOC it is easy – just set \psi to zero at the Southern boundary. But for regional MOCs, like the Atlantic, we use the second equation with v along a connected set of edges near \theta_0 (33S, say) and integrate from the bottom up. That is our southern boundary condition \Psi(z,theta_0) to then integrate northward with the first equation. For Method B, the stream function is just zero at the top or bottom.

Although it is implied, it would be useful to state in each figure caption whether you are using Method A or B.

Grammatical comments Please change MPAS-o to MPAS-Ocean throughout. p.1: original conversation, which recent years, considerable double )): 2019) and 2009) p.2 easy-to-implement of the dual mesh differ from that p.3 faces of the scalar control of the edge in the model, however care same way as in the model if fluxes are to be

properly analysed. and, we would guess, generally known. p. 4 In these definitions southern latitude, in radians, because the full velocity The computational procedure is straightforward p.5 in more general cases. p.6 Using the coarsest bin p.10 The general idea of the simple We emphasize that the algorithms described are still sensitive to parameter choices, for in each case

Please also note the supplement to this comment:
https://www.geosci-model-dev-discuss.net/gmd-2019-336/gmd-2019-336-SC1-supplement.pdf

---

## Referee Comment (RC1) · Anonymous Referee #1 · 19 May 2020

Comment on "Simple algorithms to compute meridional overturning and barotropic streamfunction on unstructured meshes"

This paper clearly explains how to calculate the meridional overturning and the barotropic streamfunction on unstructured meshes. The method proposed avoids interpolation (which to my knowledge is the most common procedure) and preserves conservation. The paper is well written, and clearly discusses the mathematics of the computation, and the sensitivity of the parameters. I am not aware of any other publication that discusses these calculations. The paper could basically be published as it is, and it will be a useful reference for the community of ocean modellers aiming to

reproduce this calculation.

See below for some minor comments that may improve it

I find confusing the notation of the streamfunctions on top of page 4. The first one can be confused with the multiplication of two integrations when it should read a double integral.

Equations could be numbered so that it is easier to reference to them.

When explaining introducing the ways of computing the meridional overturning, stating which is method A and which is method B would ease the reading.

Could you please also state in the figure captions whether the calculation is made method A or B?

p.1 original conservation, which

p1 L8 Over recent years, a considerable

p1 double )): L15 2019)

p1 double)) L21 2009)

p.2 L5 easy-to-implement

p2 L21 of the dual mesh differ from that

p.3 L1 faces of the scalar control of the edge

p.3 L6 however care

p.3 L6 same way as in the model if fluxes are to be properly analysed.

P.3 L10 and, we would guess, generally known.

p. 4 L1 In these definitions

p.4 L2 southern latitude, in radians,

p.4 L3 because the full velocity

p.4 L10 The computational procedure is straightforward

p.5 L5 in more general cases.

P.5 L5 Does the sentence "The method B should be used in more general cases" belong here? Or do authors refer to method A?

p.6 L7 Using the coarsest bin

p.10 L1 The general idea of the simple

p10 L10 We emphasize that the algorithms described are still sensitive to parameter choices, for in each case

---

## Author Response (AR1)

**Answers to reviewer 1**

**We thank Mark R. Petersen for reading our manuscript and for his constructive suggestions. Please find our answers below.**

**The Authors**

*This paper provides a clear explanation on how to compute the meridional overturning and barotropic streamfunction on unstructured meshes. It is well-written and clearly communicates the motivation, the mesh structure, and mathematics of the computation, and the sensitivity of the parameters. I have not seen these details discussed in other publications, so it will be a useful reference to ocean modelers reproducing this calculation. In MPAS-Ocean, we compute the meridional overturning circulation (MOC) streamfunction by integrating the vertical velocity in latitude bins, level-wise, as described in this paper as Method A. I adapted this method from our previous model, POP, which is a structured quadrilateral mesh but usually ran on a tri-pole grid, so the generality of latitude binning was required for that model as well. We have the ability to run the MOC calculation as an in-situ calculation during the simulation: https://github.com/MPAS-Dev/MPAS-Model/blob/master/ src/core_ocean/analysis_members/mpas_ocn_moc_streamfunction.F#L518 or as a post-processing calculation from the time-averaged velocity fields: https://github.com/MPAS-Dev/MPAS-Analysis/blob/develop/mpas_analysis/ocean/streamfunction_ moc.py#L436 We describe the philosophy of the in-situ computation here: Woodring, J., Petersen, M., Schmeisser, A., Patchett, J., Ahrens, J., Hagen, H., In Situ Eddy Analysis in a High-Resolution Ocean Climate Model, IEEE Transactions on Visualization and Computer Graphics, vol.22, no.1, pp.857-866, Jan. 31 2016 http://ieeexplore.ieee.org/stamp/stamp.jsp?arnumber= 7192723 If you feel it is relevant, you are free to add any of that background to your paper.*

**We thank the reviewer for informing us about similar techniques being used in MPAS-Ocean and in POP. At the time of writing we were aware only of the cubed-sphere configuration of MITgcm using a similar strategy of transport computation (Adcroft et al., 2004). In this paper we aim at documenting these techniques for unstructured meshes in order to facilitate the use of unstructured-mesh models by a broader community. We corrected the introduction mentioning MPAS-Ocean and POP and also cite Woodring et al. 2016 on in situ computation.**

*We have found that the python post-processing method is fine at lower resolutions, but at high resolutions (  1 million horizontal cells) it is extremely slow or fails, and we rely on our in-situ calculation. It would be good to hear in section 5 about any performance limitations in your calculation. On page 9 line 20, are these python routines just the plotting part, or do they reproduce the functionality of the Fortran and c code? If the later, did you run into performance bottlenecks, and how did you run python at high resolution – some combination of xarray and dask?*

**Diagnostic of MOC in z coordinate is fully done in python (method A, by integrating the vertical velocity in latitude bins, level-wise). The dask and xarray are used to read a 3D field if the mean over some time interval is required. The MOC calculation requires that the data that are located in memory. The reason is that we did not manage to make dask and xarray working efficiently with selecting "arbitrary" indices, which is required for computing vertical flux contribution from a cell and selecting cells that belong to a certain bin. Putting unused values to np.nan, when possible, instead of selecting by index, causes a good performance gain. One should have, of course, a sufficient amount of memory on the post processing machine. Our experience shows that 200G is enough to compute MOC for a mesh with  20M surface vertices. For a mesh with 1.3M surface vertices (2.6M surface cells) and 49 levels it takes about 7 seconds to compute a global MOC when using 91 latitudinal bins. For the largest mesh we have (23M vertices), and 80 levels it takes about 7 minutes. The python function can be found here: https://github.com/FESOM/pyfesom2/blob/ 7609c4ec54c13bcdbc1d15abc5da14bc40b9ec63/pyfesom2/diagnostics.py#L348 Computation of MOC in density coordinate can be also done in python in the same manner as methods A or B. For this, the meridional velocity (for method A) or horizontal divergence (for method B) need to be remapped conservatively onto density bins. This, however, can be done only in situ and results in 25% slow down of the code if 80 density classes are used. Method B can be then used straightforward in python. For method A we need first diagnose the diapycnal velocity from horizontal divergence. This**

is also done in python using xarray and dask.

Computation of the barotropic streamfunction is currently also made offline and we confirm that it is slow. Hence we plan to implement the in situ computation of the barotropic streamfunction.

As suggested by the reviewer we included this discussion into the chapter 5 of the revised paper.

*Your equations are missing numbers.*

**Fixed!**

*I find the notation of the streamfunction definition confusing (page 4, top). I prefer to see the $d\theta$ and $dz$ at the very end, to indicate what is included in the integral. The first one, in particular, could be confused with the multiplication of two integrations, rather than one case of a double integral. Another way to clarify is to write w and v as: $w(x, \theta', z)$ and $v(x, \theta, z')$ and integrate in $\theta'$ on the first equation and $z'$ on the second, to explicitly state that the primes are dummy integration variables, not to be confused with the upper bound and independent variable on the left side. For the new reader, it would be useful to review the definition of a streamfunction at the beginning of section 3. That is, if we have zonally-integrated velocities W and V, then for an incompressible fluid we can define a streamfunction $\Psi$ as $d\Psi/d\theta = W$ (1) $d\Psi/dz = -V$ (2) and refer to some fluid dynamics textbook – my favorite is Kundu, https://www.amazon.com/Fluid-Mechanics-Pijush-K-Kundu/dp/0123821002 (I forget which line is negative, I always have to work through it. I assume one of your streamfunctions has a negative sign absorbed in a coordinate or variable definition, but that would be good to note as well). From my equations above, it is clear that one may compute $\Psi$ by either integrating (1) in $\theta$, or integrating (2) in z. This is identical analytically, but leads to different implementations and numerical results, as you describe.*

**Thank you, we modified the equations according to reviewer's suggestion and revised the definition of the stream-function in section 3.**

*Please state the boundary conditions for the two equations, top of page 4. For Method A and a global MOC it is easy – just set $\psi$ to zero at the Southern boundary. But for regional MOCs, like the Atlantic, we use the second equation with v along a connected set of edges near $\theta_0$ (33S, say) and integrate from the bottom up. That is our southern boundary condition $\Psi(z, theta_0)$ to then integrate northward with the first equation. For Method B, the stream function is just zero at the top or bottom.*

**Done!**

*Although it is implied, it would be useful to state in each figure caption whether you are using Method A or B.*

**We modified the figure captions in the revised paper as suggested by the reviewer.**

*Grammatical comments, Please change MPAS-o to MPAS-Ocean throughout...*

**fixed! we have also corrected for all other grammar comments listed by the reviewer!**

**Answers to reviewer 2**

**We thank the reviewer for reading our manuscript and for giving suggestions. Please find our answers below.**

**The Authors**

*This paper clearly explains how to calculate the meridional overturning and the barotropic streamfunction on unstructured meshes. The method proposed avoids interpolation (which to my knowledge is the most common procedure) and preserves*

*conservation. The paper is well written, and clearly discusses the mathematics of the computation, and the sensitivity of the parameters. I am not aware of any other publication that discusses these calculations. The paper could basically be published as it is, and it will be a useful reference for the community of ocean modellers aiming to reproduce this calculation. See below for some minor comments that may improve it I find confusing the notation of the streamfunctions on top of page 4. The first*
5 *one can be confused with the multiplication of two integrations when it should read a double integral.*

**Thank you, we modified notations of the streamfunctions in section 3.**

*Equations could be numbered so that it is easier to reference to them.*

**Fixed!**

*When explaining introducing the ways of computing the meridional overturning, stating which is method A and which is method B would ease the reading.*

**Done!**

*Could you please also state in the figure captions whether the calculation is made method A or B?*

20 **Done! We modified the figure captions in the revised paper stating the methodology and we also corrected for all grammar comments listed by the reviewer!**

[revised manuscript text omitted]

**3 Meridional overturning**

 For zonally-integrated vertical and meridional velocities $W = \int_{x_w}^{x_e} w(x', \theta, z) dx'$ and $V = \int_{x_w}^{x_e} v(x', \theta, z) dx'$ of a divergence-less flow we can introduce (see eg. Kundu et al., 2012) a streamfunction $\Psi(z, \theta)$

$$\Psi(z, \theta) = \int_{\theta_s}^{\theta} R_E d\theta \int_{x_w}^{x_e} w \, dx,$$

$$\Psi(z, \theta) = \int_{-H}^{z} dz \int_{x_w}^{x_e} v \, dx.$$

 such that

$$\frac{1}{R_E} \frac{\partial \Psi}{\partial \theta} = W, \qquad \frac{\partial \Psi}{\partial z} = -V. \tag{2}$$

Here $\theta$ is the latitude in radians, $z$ is the  depth, $R_E$ is Earth's radius, $v$ and $w$ are the meridional and vertical velocities and $x_w$ and $x_e$ the western and eastern boundaries in zonal direction. Following the definition, there are two convenient ways of computing global $\Psi$ in geopotential coordinates:

$$\Psi(z,\theta) = \Psi(z,\theta_r) + \int_{\theta_r}^{\theta} R_E \int_{x_w}^{x_e} w(x',\theta',z)d\theta'dx', \tag{3}$$

or

$$\Psi(z,\theta) = -\int_{-H}^{z} \int_{x_w}^{x_e} v(x',\theta,z')dz'dx'. \tag{4}$$

Here $\theta_r$ the reference latitude. For global MOC computations it is the southernmost latitude of the Antarctic coast, where $\Psi(z,\theta_r)=0$. For regional MOCs, like the Atlantic MOC (AMOC), $\theta_r$ is any convenient latitude where $\Psi(z,\theta_r)=0$ should be provided. For this, the equation 4 is usually used. In this equation the boundary condition is naturally taken into account by integrating from the bottom $z=-H(x,\theta)$. Note that both ways of computation are equivalent because the full velocity vector is divergence-free. In the following we discuss details of both methods of computation on unstructured meshes. Method A (equation 3) involves vertical velocities and is more straightforward. Method B (equation 4) is based on horizontal velocities and is slightly more complicated.

[revised manuscript text omitted]

actual line through which transport is computed is a broken line composed of vectors $(\mathbf{d}_{ec_1}, \mathbf{d}_{ec_2})$ related to the crossed edges. For a triangular C grid discretization one will deal with transports directly through the edges. The caveat in this case is that some of the crossed edges will be hanging and not contributing to the broken line. They are excluded by noticing that they have vertices that are encountered only once in the union of vertices of crossed edges. On hexagonal C grids the procedure needs to deal with edges of dual triangular mesh. We denote the set of edges forming the broken line around $\theta = \theta_i$ as $e(i)$.

– The flux associated to the edge is given by the expression for $F_e$ above. The question now is the orientation of edges. This question is trivially solved for each $e$ by taking $F_e$ if $\theta_{v_1} - \theta_i > 0$ and $-F_e$ otherwise. It corresponds to keeping the normals to segments oriented so that transports are from the "northern" side of the broken curve. On triangular C grid the edge normal vectors used to introduce edge velocities can be selected as turned $90°$ in positive direction from the edge direction. This will allow to solve the orientation problem in the same way.

– Since each of segments $(\mathbf{d}_{ec_1}, \mathbf{d}_{ec_2})$ belongs to a particular cell, vertical integration is trivial for fixed level surfaces. If level surfaces are moving, the fluxes (transports) through the faces associated with segments are conservatively interpolated to the desired system of levels assuming linear distribution within model layers. In particular, the new system of

levels can be specified in terms of potential density, with the result being the streamfunction in density coordinates. For each level the contributions from edges $e \in e(i)$ are summed to get streamfunction at this level and the latitude $\theta_i$.

Note that the set of intersected edges may be ordered arbitrarily, the computation relies on the orientation of edges with respect to lines $\theta = \theta_i$. This is the reason why the search for intersected edges remains relatively fast even on very large meshes.

5    Furthermore, it needs to be done only once for a particular mesh. Similarly to Method A, computations can be generalized to any set of lines, in particular to isolines of mean sea surface height or barotropic streamfunction. In both Methods we introduce masks if computations need to be confined to a particular basin.

Using this method we computed the streamfunction using the discrete spacing of $\Delta\theta = 0.125\,°$. The difference to the streamfunction computed by method A is illustrated in Fig. 7. The discrepancy between both methods is caused by the difference of

10   attribution of ocean volume to $\theta_i$. This, as shown in Fig. 7, can lead to a differences exceeding locally 1 Sv. These differences are not the errors, but uncertainty in the interpretation (see further).

 If the modeled fluxes have been remapped onto the desired set of vertical levels as, for instance prescribed density levels, method B can be directly used for computing the MOC for a new  vertical coordinate system. Figure 5 depicts the MOC computed using $\sigma_2$ (density

15   referenced to 2000m) coordinate in vertical. For computing the streamfunction in density coordinate we used 1000 equally spaced $\sigma_2$ levels varying from 1027.5 to 1037.5 $kg/m^3$. The resulting MOC resembles that of generally known pattern from literature, with less expressed Deacon cell  relative $z$ coordinate  streamfunction. The result is sensitive to the selection of density bins, as illustrated in the bottom panel of Fig. 5 where the difference is presented with computations relying on the density levels of Megann (2018). He used 72 unequally spaced density classes spanning the range $30.0 < \sigma_2 <$

20   $37.2\ kg/m^3$ and using the logarithmic scale for densities higher than $\sigma_2 > 35.0\ kg/m^3$ to better represent the deep and bottom waters. Thus, due to the different sampling the difference in the equatorial overturning of the surface waters reaches ~3Sv for $30 < \sigma_2 < 35.0\ kg/m^3$ and is even larger for the circulation cell associated with the Antarctic Bottom Water. We conclude that different or not detailed enough selection of density levels may result in the small-scale recirculations in  the diagnosed MOC. However, this difference is not an error but attribution uncertainty created by arbitrariness in the selection of

25   density levels.

Note, that diagnostic of MOC in density coordinate can be also made in the same manner as method A. For this, the horizontal divergence needs to be remapped conservatively into density bins. From the horizontal divergence we then (1) diagnose the diapycnal velocity and (2) use it in method A.

[revised manuscript text omitted]
 either vertical or horizontal velocities to be stored as $w_{k,v}$ or $(u,v)_{k,c}$ where $k$, $v$ and $c$ denote the layer and element indices, vertex or element indices, respectively. This is the default output provided by FESOM. For computing the barotropic streamfunction and MOC in density space the index $k$ refers to a density bin and $w_{k,v}$ (is then diagnosed from the horizontal divergence within the bins) or $(u,v)_{k,c}$ denotes the transport through this bin below the element $c$. Transports within the density classes are instantaneously computed by FESOM and stored with the desired frequency if option $ldiag\_dMOC$ is activated. For the sake of better subsampling, the number of density classes for computing transports shall be sufficiently large. This, however, can make the remapping of

transports onto density bins computationally heavy. Our experience shows that instantaneous remapping of modeled fluxes onto density classes results in a ∼ 25 % slow down of the code if 80 density classes are used. For this reason $ldiag\_dMOC$ is switched off per default.

For postprocessing in python a combination of Dask and Xarray is used for reading a 3D field (if, for example, the mean over several timesteps or years is required). The MOC calculation itself happens on the data that are located in memory. One should have, of course, a sufficient amount of memory on the post processing machine. Our experience shows that 200G is enough to compute MOC for a mesh with ∼ 23 M surface vertices. For a mesh with ∼ 1.3M surface vertices and 49 vertical levels it takes about 7 seconds to compute a global MOC using 91 latitudinal bins. For the largest mesh we have tested in FESOM so far (23M surface vertices), and 80 levels in vertical same computation takes about 7 minutes.

Computation of the barotropic streamfunction is currently implemented offline and from our experience it is slow because of the loops along vertical and zonal directions are required. Hence we plan to implement the computation of the barotropic streamfunction following the philosophy of the in-situ computations (see e.g. Woodring et al., 2015).

**6 Discussion**

[revised manuscript text omitted]

**Figure 6.** Schematics of edge search method. The gray line $L$ intersects edges depicted with arrows that show their orientation. The set of segments drawn to centroids from the centers of intersected edges forms a broken line connecting land at left to land at right where exact expressions for fluxes are available in FESOM2. The broken line formed by the intersected edges will be taken on triangular C grids, and on hexagonal C grids it will be composed of edges of primary hexagonal mesh. The set of intersected edges may stay disordered, only edge orientation with respect to the line $L$ should be known. The latter is positive if the latitude of the first edge point is larger than that of $L$ and negative otherwise. The transport through $L$ is the transport through the associated broken line.

[Figure]

**Figure 7.** Differences between computations of MOC using meridional (method B) or vertical (method A) velocities. The discrepancy between techniques may result in differences of more than 1 Sv.

[Figure]

**Figure 8.** Uppper panel: the barotropic streamfunction computed using $\Delta\theta, \Delta\phi = 0.25°$. Middle and bottom panels show the differences in cases $\Delta\theta = 1°$ and $2°$, respectively.